# Breakdown of magnons in a strongly spin-orbital coupled magnet

Stephen M. Winter [1], Kira Riedl[1], Pavel A. Maksimov[2], Alexander L. Chernyshev[2], Andreas Honecker [3] & Roser Valentí[1]

The description of quantized collective excitations stands as a landmark in the quantum theory of condensed matter. A prominent example occurs in conventional magnets, which support bosonic magnons—quantized harmonic fluctuations of the ordered spins. In striking contrast is the recent discovery that strongly spin-orbital-coupled magnets, such as $\alpha$-$RuCl_3$, may display a broad excitation continuum inconsistent with conventional magnons. Due to incomplete knowledge of the underlying interactions unraveling the nature of this continuum remains challenging. The most discussed explanation refers to a coherent continuum of fractional excitations analogous to the celebrated Kitaev spin liquid. Here, we present a more general scenario. We propose that the observed continuum represents incoherent excitations originating from strong magnetic anharmonicity that naturally occurs in such materials. This scenario fully explains the observed inelastic magnetic response of $\alpha$-$RuCl_3$ and reveals the presence of nontrivial excitations in such materials extending well beyond the Kitaev state.

[1] Institut für Theoretische Physik, Goethe-Universität Frankfurt, Max-von-Laue-Strasse 1, 60438 Frankfurt am Main, Germany. [2] Department of Physics and Astronomy, University of California, Irvine, CA 92697, USA. [3] Laboratoire de Physique Théorique et Modélisation, CNRS UMR 8089, Université de Cergy-Pontoise, 95302 Cergy-Pontoise Cedex, France. Correspondence and requests for materials should be addressed to S.M.W. (email: winter@physik.uni-frankfurt.de)

From magnons in ordered magnets to phonons in periodic crystals, the appearance of bosonic collective excitations is ubiquitous in condensed phases of matter[1]. For this reason, special attention is given to those states that support more exotic collective modes, for which the conventional paradigm breaks down. In the context of magnetic phases, the breakdown of magnons is commonly thought to require closeness to an unconventional state such as a quantum spin liquid[2–4]. A notable example occurs in Kitaev's exactly solvable honeycomb model[5], for which strongly anisotropic and bond-dependent interactions fractionalize conventional spin excitations into Majorana spinons and fluxes. This Kitaev state has recently risen to prominence due to the suggestion that it may be realized in heavy metal $4d^5$ and $5d^5$ insulators via a specific interplay between the crystal field and strong spin-orbit coupling[6], and, consequently, a variety of candidate materials based on $Ir^{4+}$ and $Ru^{3+}$ have been intensively explored[7]. Encouragingly, evidence of a continuum of magnetic excitations that is inconsistent with conventional magnons was found in the majority of such materials, including the two-dimensional (2D) honeycomb $Na_2IrO_3$[8, 9] and $\alpha$-$RuCl_3$[10–14], as well as the three-dimensional (3D) analogs $\beta$-,$\gamma$-$Li_2IrO_3$[15], despite all of them having magnetically ordered ground states.

While the observed excitation continua in these systems have been interpreted in terms of signatures of the Kitaev state, the low-symmetry crystalline environment of the real materials also allows various additional interactions beyond Kitaev's model[16–18], which are thought to be large based on both experimental[19, 20] and theoretical[18, 21, 22] considerations. In this sense, understanding the mechanism for the breakdown of magnons and the appearance of a broad continuum of magnetic excitations remain a key challenge.

In this work, we study a representative case $\alpha$-$RuCl_3$, which forms a layered 2D honeycomb lattice and displays zigzag magnetic order below $T_N \sim 7\,K$[12, 13, 23]. We specifically address the recent inelastic neutron scattering (INS) measurements, which have revealed low-energy magnons[24] coexisting with an intense excitation continuum[12]. The latter continuum possesses a distinctive six-fold star shape in momentum space, and large intensity at the 2D $\Gamma$-point over a wide energy range $E = 2$–$15$ meV[12]. To resolve the nature of this continuum, we take two complementary approaches. We first theoretically investigate the neutron spectra over a range of relevant magnetic interactions in order to determine the correct spin Hamiltonian for $\alpha$-$RuCl_3$, which has been a subject of intense recent discussion[18, 25–28]. Second, we identify the conditions that lead to the breakdown of conventional magnons in the presence of strongly anisotropic and frustrated interactions, revealing that nontrivial excitations naturally persist well beyond the Kitaev spin liquid.

## Results

**The model.** Based on previous ab initio studies[18, 25–28], the largest terms in the spin Hamiltonian of $\alpha$-$RuCl_3$ are generally expected to include nearest neighbor Heisenberg $J_1$, Kitaev $K_1$, and off-diagonal $\Gamma_1$ couplings, supplemented by a third neighbor Heisenberg $J_3$ term:

$$\mathcal{H} = \sum_{\langle i,j \rangle} J_1\, \mathbf{S}_i \cdot \mathbf{S}_j + K_1 S_i^\gamma S_j^\gamma + \Gamma_1 \left( S_i^\alpha S_j^\beta + S_i^\beta S_j^\alpha \right) + \sum_{\langle\langle\langle i,j \rangle\rangle\rangle} J_3\, \mathbf{S}_i \cdot \mathbf{S}_j , \quad (1)$$

where $\langle i,j \rangle$ and $\langle\langle\langle i,j \rangle\rangle\rangle$ refer to summation over first and third neighbor bonds, respectively (see Fig. 1). The bond-dependent variables $\{\alpha, \beta, \gamma\}$ distinguish the three types of first neighbor bonds, with $\{\alpha, \beta, \gamma\} = \{y, z, x\}$, $\{z, x, y\}$, and $\{x, y, z\}$ for the X-bonds, Y-bonds, and Z-bonds, respectively. The third neighbor

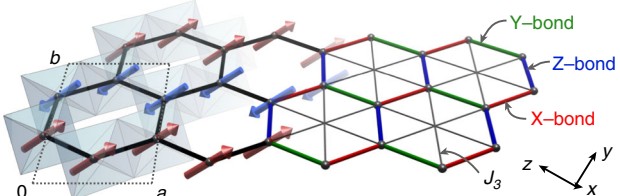

**Fig. 1** From material to model. Within the honeycomb $ab$-layer of $\alpha$-$RuCl_3$ are illustrated the $RuCl_6$ octahedra, magnetic zigzag ordering pattern, and definition of the underlying magnetic interactions. Crystal axes are labeled with respect to the $C2/m$ structure

interactions are bond-independent. The phase diagram of this model has been discussed previously[17, 18, 26, 29], and is further detailed in Supplementary Note 1; here we review the key aspects.

In the limit $J_1 = \Gamma_1 = J_3 = 0$, the ground state is a gapless $Z_2$ spin liquid for either positive or negative $K_1$, as demonstrated in Kitaev's seminal work[5]. Small perturbations from the pure $K_1$ limit may induce various magnetically ordered states, such as the zigzag antiferromagnetic (AFM) state observed in $\alpha$-$RuCl_3$ and shown in Fig. 1. The simplest perturbation is the introduction of a finite $J_1$, which yields the well-studied $(J_1, K_1)$ nearest neighbor Heisenberg-Kitaev (nnHK) model. This model hosts zigzag order in the region $K_1 > 0$, $J_1 < 0$, as discussed in Supplementary Note 1. Accordingly, previous analysis of the powder INS experiments within the context of the nnHK model[13] suggested that $K_1 \sim +7$ meV, and $|J_1/K_1| \sim 0.3 - 0.7$ for $\alpha$-$RuCl_3$. On this basis, the excitation continua observed experimentally were initially interpreted in terms of proximity to the AFM $K_1 > 0$ spin liquid[12, 13]. However, the further consideration of finite $\Gamma_1$ and $J_3$ interactions in Eq. (1) significantly expands the experimentally relevant region, as both interactions generally stabilize zigzag order. Indeed, recent ab initio studies[18, 25–28] have suggested that the zigzag order in $\alpha$-$RuCl_3$ emerges from $J_1 \sim 0$, $K_1 < 0$, $\Gamma_1 > 0$, and $J_3 > 0$, with $|\Gamma_1/K_1| \sim 0.5 - 1.0$ and $|J_3/K_1| \sim 0.1 - 0.5$, as reviewed in Supplementary Note 2. That is, $K_1$ is ferromagnetic, and supplemented by significant $\Gamma_1$ and $J_3$ interactions. Such interactions would represent large deviations from both Kitaev's original model and the region identified by initial experimental analysis. Before discussing the origin of the excitation continua, it is therefore crucial to first pinpoint the relevant interaction parameters.

In order to address this issue directly, we have computed the neutron scattering intensity $\mathcal{I}(\mathbf{k}, \omega)$ for a variety of interactions within the zigzag ordered phase via both linear spin-wave theory (LSWT) and exact diagonalization (ED). For the latter case, we combine results from various periodic 20-site and 24-site clusters compatible with the zigzag state in order to probe a wider range of $\mathbf{k}$-points (see "Methods" section). Full results for the complete range of models are presented in Supplementary Note 5. Here, we highlight the key results for two representative sets of interactions. Within the $(J_1, K_1)$ nnHK model, we focus on Model 1 ($J_1 = -2.2$, $K_1 = +7.4$ meV; $|J_1/K_1| = 0.3$), which lies on the border of the region initially identified in ref. [12], close to the spin liquid. Beyond the nnHK model, we consider Model 2 ($J_1 = -0.5$, $K_1 = -5.0$, $\Gamma_1 = +2.5$, $J_3 = +0.5$ meV) for which parameters have been guided by recent ab initio studies[18, 25–28], and further optimized to improve agreement with the experimental spectra. Results for Models 1 and 2 are first presented in Figs. 2 and 3, which show detailed $\omega$-dependence and $\mathbf{k}$-dependence of $\mathcal{I}(\mathbf{k}, \omega)$, along with the evolution of the spectra upon changing parameters toward the $K_1 > 0$ or $K_1 < 0$ spin liquid regions.

**Nearest neighbor Heisenberg-Kitaev model.** We begin by analyzing the spectra $\mathcal{I}(\mathbf{k}, \omega)$ within the zigzag phase of the $(J_1, K_1)$

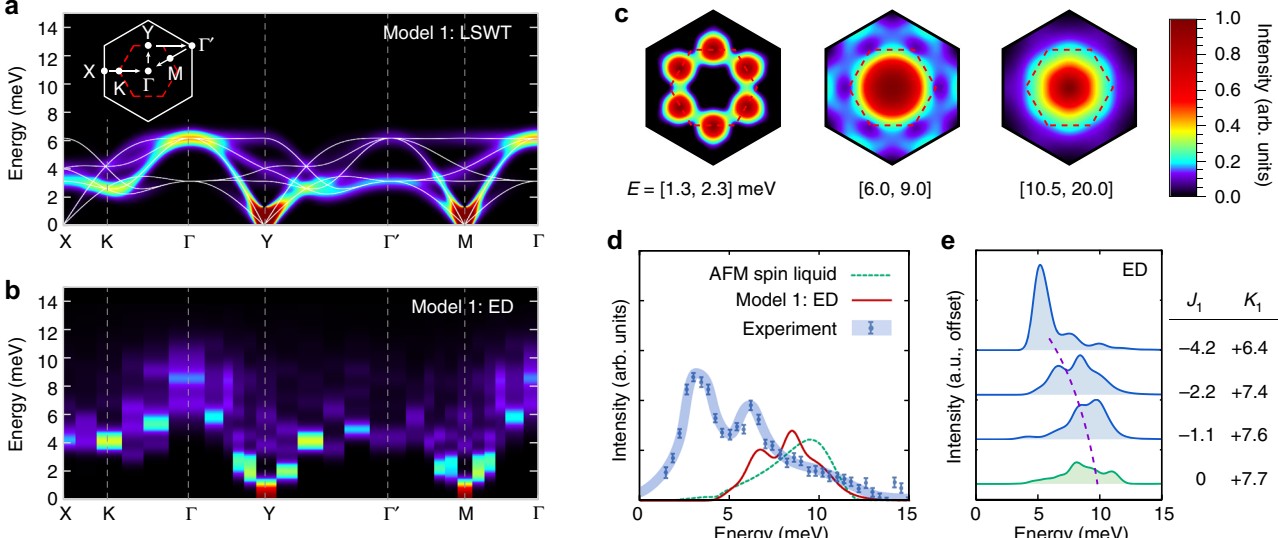

**Fig. 2** Neutron scattering intensity $\mathcal{I}(\mathbf{k}, \omega)$ within the nnHK model. **a–c** Detailed results for Model 1 ($J_1 = -2.2$, $K_1 = +7.4$ meV): **a** $\mathcal{I}(\mathbf{k}, \omega)$ computed via LSWT; results are averaged over the three zigzag ordering wavevectors, parallel to the X-bonds, Y-bonds, and Z-bonds. Inset: Definition of Brillouin zone and high-symmetry $\mathbf{k}$-points. **b** ED results, combining data from several 20-site and 24-site periodic clusters (see "Methods"). **c** ED $\mathbf{k}$-dependence of $\mathcal{I}(\mathbf{k}, \omega)$ integrated over the indicated energies, as obtained from a single 24-site cluster respecting all symmetries of Eq. (1) (see "Methods"). **d** Comparison of $\Gamma$-point intensities for the $K_1 = +7.7$ meV AFM spin liquid (exact results[53, 54]), Model 1 (ED), and the experimental data for $\alpha$-RuCl$_3$[12]. **e** Evolution of the ED $\Gamma$-point intensity with decreasing $|J_1/K_1|$, showing absence of low-energy intensity close to the $K_1 > 0$ spin liquid. The top three interaction sets correspond to zigzag order, while the bottom is the $K_1 > 0$ spin liquid. For all spectra, a Gaussian broadening of 0.5 meV has been applied

nnHK model, starting with Model 1 (Fig. 2). Despite proximity to the spin liquid, the ED calculations on Model 1 (Fig. 2b) show sharp dispersive modes appearing over the majority of the Brillouin zone that are consistent with the conventional magnons of LSWT (Fig. 2a). Indeed, for energies below the main spin-wave branch ($\omega = 1.3$–$2.3$ meV), intensity is localized around the M-points and Y-points, corresponding to the pseudo-Goldstone modes of the zigzag order (Fig. 2c). ED calculations show clear spin-wave cones emerging from such points and extending to higher energies. Large deviations from LSWT are observed only for the highest energy excitations, which appear near the 2D $\Gamma$-point for energies $\omega > 5$ meV. Here, the ED calculations display a broad continuum-like feature centered at $\omega \sim K_1$ that resembles the response expected for the $K_1 > 0$ Kitaev spin liquid, as shown in Fig. 2d. However, comparison with the experimental $\mathcal{I}(\Gamma, \omega)$ shows poor agreement, while the experimental intensity extends from 2 to 15 meV, the ED results for Model 1 predict intensity only at high energies $>5$ meV. Indeed, the evolution of the $\Gamma$-point intensity with $|J_1/K_1|$ is shown in Fig. 2e. On approaching the $K_1 > 0$ spin liquid by decreasing $|J_1/K_1|$, excitations at the $\Gamma$-point shift to higher energy, such that none of the parameters in the vicinity of the spin liquid reproduce the experimental $\omega$-dependence of $\mathcal{I}(\Gamma, \omega)$. Similar conclusions can also be drawn from recent Density Matrix Renormalization Group (DMRG) studies of the nnHK model[30]. We therefore conclude that the broad features observed experimentally in $\mathcal{I}(\Gamma, \omega)$ at relatively low energies[12] are incompatible with the nnHK model with $J_1 < 0$ and $K_1 > 0$.

**Extended ab initio guided model.** In order to treat the effect of interactions beyond the nnHK model, we consider now the ab initio guided Model 2. In contrast to Model 1, ED calculations on Model 2 (Fig. 3b) show large deviations from standard LSWT (Fig. 3a) over a wide range of $\mathbf{k}$ and $\omega$. This model reproduces many of the experimental spectral features[12, 24]. In particular, sharp single-magnon-like peaks appear only near the pseudo-Goldstone modes at the M-points and Y-points. Elsewhere in the

Brillouin zone, broad continuum-like features are observed within the ED resolution. As demonstrated in Fig. 3c, we find significant intensity at low energies ($\omega < 2.3$ meV), at both the $\Gamma$-points and (M,Y)-points. For the intermediate energy region ($\omega = 5.5$–$8.5$ meV), $\mathcal{I}(\mathbf{k})$ resembles the six-fold star shape observed in ref. [12]. At higher energies ($\omega > 10.5$ meV) scattering intensity is mainly located at the $\Gamma$-point, also in accord with experiment. Furthermore, the ED results for the $\Gamma$-point intensity $\mathcal{I}(\Gamma, \omega)$ show a broad range of excitations peaked around 4 and 6 meV, and extending up to ~15 meV (Fig. 3d). Therefore, ED calculations on Model 2 reproduce all of the main experimental spectral features, validating the range of interactions indicated by ab initio calculations. The only aspect that is not quantitatively reproduced within the Model 2 is the magnitude of the gap at the M-point (~0.8 meV at the level of LSWT vs. ~2 meV experimentally[13, 24]). This discrepancy may result from deviations from $C_3$ symmetry, which are allowed within the $C2/m$ space group[18, 31], but not considered here for simplicity (see Supplementary Fig. 11). Interestingly, the spectral features at the $\Gamma$-point become dramatically sharper on approaching the $K_1 < 0$ spin liquid, as shown in the evolution of $\mathcal{I}(\Gamma, \omega)$ with the ratio $|\Gamma_1/K_1|$ (Fig. 3e). This result reveals that the broad continuum may not be directly associated with a proximity to the Kitaev state.

**Magnon stability beyond LSWT.** To gain further insight into the reason for such a drastic contrast between the stability of magnons in Models 1 and 2, it is useful to consider possible magnon decay channels in the zigzag ordered phase. At the level of LSWT, the spin-wave Hamiltonian is truncated at quadratic order, and can be written $\mathcal{H}_2 = \sum_{\mathbf{k},m} \epsilon_{\mathbf{k},m} a^{\dagger}_{\mathbf{k},m} a_{\mathbf{k},m}$ in terms of magnon creation (annihilation) operators $a^{\dagger}$ ($a$), where $\epsilon_{\mathbf{k},m}$ denotes the dispersion of the $m$th magnon band. In this harmonic approximation, magnons represent sharp, well-defined excitations. However, when higher-order anharmonic terms are included, the total magnon number $N_{\text{tot}} = \sum_{\mathbf{k},m} a^{\dagger}_{\mathbf{k},m} a_{\mathbf{k},m}$ is typically not a conserved quantity, such that the stability of magnons is not guaranteed beyond quadratic order. Quantum

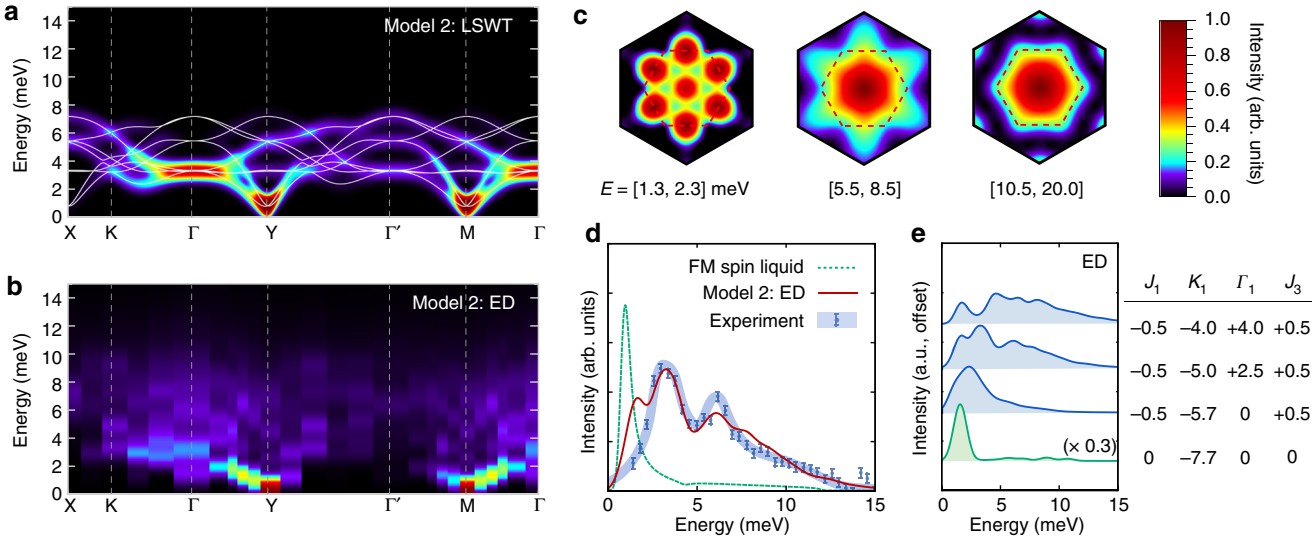

**Fig. 3** Neutron scattering intensity $\mathcal{I}(\mathbf{k}, \omega)$ within the extended model. **a–c** Detailed results for Model 2 ($J_1 = -0.5$, $K_1 = -5.0$, $\Gamma_1 = +2.5$, $J_3 = +0.5$ meV): **a** $\mathcal{I}(\mathbf{k}, \omega)$ computed via LSWT; results are averaged over the three zigzag domains with ordering wavectors parallel to the X-bonds, Y-bonds, and Z-bonds. **b** ED results, combining data from several 20-site and 24-site periodic clusters (see "Methods"). **c** ED $\mathbf{k}$-dependence of $\mathcal{I}(\mathbf{k}, \omega)$ integrated over the indicated energies, as obtained from a single 24-site cluster respecting all symmetries of Eq. (1) (see "Methods"). **d** Comparison of $\Gamma$-point intensities for the $K_1 = -7.7$ meV FM spin liquid (exact results[53, 54]), Model 2 (ED), and the experimental data for $\alpha$-RuCl$_3$[12]. **e** Evolution of the ED $\Gamma$-point intensity with decreasing $|\Gamma_1/K_1|$, showing significant broadening at finite $\Gamma_1$. The top three interaction sets correspond to zigzag order, while the bottom is the $K_1 < 0$ spin liquid. For all spectra, a Gaussian broadening of 0.5 meV has been applied

fluctuations associated with the higher-order anharmonic decay terms may mix sharp single-magnon modes with the multi-magnon continuum[32–34]. Similar considerations also apply to the breakdown of other collective modes, such as phonons in anharmonic crystals[35, 36]. From this perspective, a large decay rate is expected for any single-magnon mode that is energetically degenerate with the multi-particle continuum, unless there are specific symmetries guaranteeing that the two do not couple. It is therefore useful to consider the prerequisites for magnon breakdown in the presence of the strongly anisotropic interactions of Eq. (1).

**Magnon decay channels for the nnHK model.** We first examine the stability of magnons in the nnHK model. For pure $J_1$ and $K_1$ interactions, the total spin projections $S_{\text{tot}}^\gamma = \sum_i S_i^\gamma$ are conserved along the cubic axes $\gamma = \{x, y, z\}$ modulo two. Since the ordered moment also lies along one of the cubic axes in the zigzag phase[20, 37] (see Fig. 4c), the possible magnon decay channels are restricted. In the local picture, the relevant quantum fluctuations are local singlet $S_i^x S_j^x |{\uparrow}{\downarrow}\rangle = |{\downarrow}{\uparrow}\rangle$ and triplet $S_i^x S_j^x |{\uparrow}{\uparrow}\rangle = |{\downarrow}{\downarrow}\rangle$ fluctuations shown in Fig. 4a, with $\Delta S_{\text{tot}}^z = 0$ and 2, respectively. In the magnon picture, the Hamiltonian can only contain even-order terms (i.e., $\mathcal{H} = \mathcal{H}_2 + \mathcal{H}_4 + \ldots$), analogous to conventional Heisenberg antiferromagnets with collinear ordered spins[32, 34]. For example, the fourth-order decay process due to $\mathcal{H}_4$ mixes the one-magnon states with the three-magnon continuum ($\Delta N_{\text{tot}} = \pm 2$), where

$$\mathcal{H}_4 = \sum_{\mathbf{1}-\mathbf{4}} V_{\mathbf{123}}^4 \, a_{\mathbf{1}}^\dagger a_{\mathbf{2}}^\dagger a_{\mathbf{3}}^\dagger a_{\mathbf{4}} \, \delta(\mathbf{k}_1 + \mathbf{k}_2 + \mathbf{k}_3 - \mathbf{k}_4) + \text{H.c.} \quad (2)$$

Here, the bold index ($\mathbf{n} \equiv \mathbf{k}_n, m_n$) labels both momentum and magnon band. This process is pictured in Fig. 4b. As noted above, the effect of such terms depends crucially on the availability of low-energy three-magnon states in which to decay.

The density of three-magnon states for Model 1 is shown in Fig. 4d, based on the one-magnon dispersions obtained in LSWT. At each $\mathbf{k}$-point, the lowest energy three-magnon state

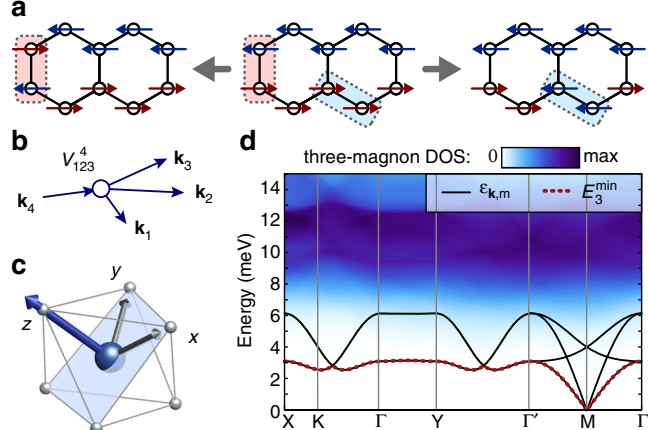

**Fig. 4** Magnon decay channels for the nnHK model. **a** Local picture of quantum fluctuations away from zigzag order. The energy cost for the left process vanishes on approaching the spin liquid $|J_1/K_1| \to 0$. **b** Momentum space picture for the corresponding fourth-order decay process due to $\mathcal{H}_4$. **c** Ordered moment direction for Model 1 ($J_1 = -2.2$, $K_1 = +7.4$ meV), corresponding to the zigzag domain with ordering wavevector $\mathbf{Q} = \text{Y}$. **d** LSWT dispersions $\epsilon_{\mathbf{k},m}$, and three-magnon density of states (DOS) for Model 1 for the same zigzag domain as **c**. The dashed line indicates the bottom of the three-magnon continuum ($E_3^{\text{min}}$), which is coincident with the lowest magnon band

$a_{\mathbf{q}_1}^\dagger a_{\mathbf{q}_2}^\dagger a_{\mathbf{q}_3}^\dagger |0\rangle$ (with $\mathbf{q}_1 + \mathbf{q}_2 + \mathbf{q}_3 = \mathbf{k}$) is obtained by placing two particles in the pseudo-Goldstone modes at opposite M-points ($\mathbf{q}_1 + \mathbf{q}_2 = 0$), and the third particle at $\mathbf{q}_3 = \mathbf{k}$, with total energy $E_3^{\text{min}}(\mathbf{k}) = \epsilon_{\mathbf{k},1} + 2\epsilon_{\text{M},1}$. This implies $E_3^{\text{min}}(\mathbf{k}) \geq \epsilon_{\mathbf{k},1}$. That is, the three-magnon states lie above the lowest one-magnon band at every $\mathbf{k}$-point. As a result, every magnon in the lowest band remains kinetically stable, due to the absence of low-energy three-particle states in which to decay. Precisely this condition ensures the stability of low-energy magnons in conventional isotropic antiferromagnets, and explains the sharp magnon-like peaks

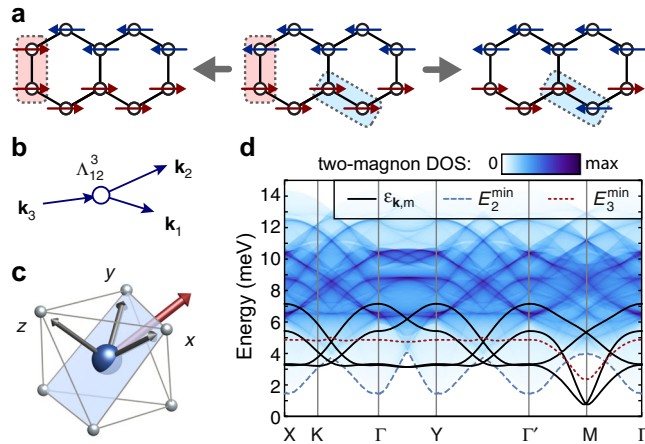

**Fig. 5** Magnon decay channels for the extended model. **a** Local picture of additional quantum fluctuations away from zigzag order induced by $\Gamma_1$ interactions. **b** Momentum space picture of the third-order decay process $\mathcal{H}_3$. **c** Ordered moment direction for Model 2 ($J_1 = -0.5$, $K_1 = -5.0$, $\Gamma_1 = +2.5$, $J_3 = +0.5$ meV) with zigzag ordering wavevector $\mathbf{Q} = Y$, parallel to the Z-bond. **d** LSWT dispersions $\epsilon_{\mathbf{k},m}$, and two-magnon DOS for Model 2 with $\mathbf{Q} = Y$. Dashed lines indicate the bottom of the two-magnon and three-magnon continuum ($E_2^{\min}(\mathbf{k})$ and $E_3^{\min}(\mathbf{k})$, respectively)

observed in Fig. 2b for Model 1. Strong spectral broadening in the nnHK model can occur only for high-lying excitations with $\epsilon_{\mathbf{k},m} > \epsilon_{\mathbf{k},1} = E_3^{\min}$, where the density of three-magnon states is finite, such as at the 2D $\Gamma$-point. On approaching the spin liquid (at $J_1/K_1 = 0$), this condition is relaxed due to the vanishing dispersion of the lowest magnon band (i.e., $\epsilon_{\mathbf{k},1} \to 0$), which corresponds to a vanishing energy cost the singlet fluctuations shown on the left of Fig. 4a. The relevant fluctuations in the limit $J_1/K_1 \to 0$ therefore correspond to $\Delta N_{\text{tot}} = \pm 2$. For other values of $J_1/K_1$, the majority of magnons are expected to remain stable due to the absence of low-energy three-magnon states.

**Magnon decay channels for the extended model.** In Model 2, the character of the quantum fluctuations away from zigzag order is notably different (Fig. 5). The finite $\Gamma_1$ interaction reduces the local symmetry and leads to rotation of the ordered moments away from the cubic axes[20, 37] (Fig. 5c). In the local picture, this allows additional single-spin fluctuations $S_i^x S_j^z |{\uparrow}{\uparrow}\rangle = |{\downarrow}{\uparrow}\rangle$ (Fig. 5a), which correspond to odd-order anharmonic terms $\mathcal{H}_3, \mathcal{H}_5, \ldots$ in the magnon Hamiltonian, where[33, 34]:

$$\mathcal{H}_3 = \sum_{1-3} \Lambda_{12}^3 \, a_1^\dagger a_2^\dagger a_3 \, \delta(\mathbf{k}_1 + \mathbf{k}_2 - \mathbf{k}_3) + \text{H.c.} \qquad (3)$$

At lowest order, such terms mix the single-magnon states with the two-magnon continuum ($\Delta N_{\text{tot}} = \pm 1$), via the scattering process depicted in Fig. 5b. The density of two-magnon states is shown in Fig. 5d, for the zigzag domain with $\mathbf{Q} = Y$. In this case, at each $\mathbf{k}$-point the lowest energy two-magnon state $a_{\mathbf{q}_1}^\dagger a_{\mathbf{q}_2}^\dagger |0\rangle$ is obtained by placing one particle in the pseudo-Goldstone mode at an M-point, and the second particle at $\mathbf{q}_2 = \mathbf{k} - M$, with total energy $E_2^{\min}(\mathbf{k}) = \epsilon_{\mathbf{k}-M} + \epsilon_M \neq E_3^{\min}$. It should be emphasized that this condition differs from that of a conventional Heisenberg antiferromagnet (for which $E_2^{\min} = E_3^{\min}$)[34]. In the case of Model 2, the difference is directly related to the strong anisotropic $K_1$ and $\Gamma_1$ interactions, which shift the pseudo-Goldstone modes to the M-points, such that only high-energy magnons remain at the $\Gamma$-point or ordering wavevector $\mathbf{Q}$[38]. This shift therefore leads to an offset of the low-energy even and odd magnon states in

$\mathbf{k}$-space such that $E_2^{\min}(\mathbf{k}) < \epsilon_{\mathbf{k},1}$ over a wide region of the Brillouin zone; there are many two-magnon states with equal or lower energy than the one-magnon states. Provided there is a finite $\Gamma_1$, the spontaneous decay of single magnons into the two-particle continuum is therefore allowed even for the lowest magnon band. The decay rate is expected to be particularly large near the zone center, which represents a minimum in $E_2^{\min}$. Similar kinematic conditions may also occur in other systems[34, 39]. For Model 2, the pseudo-Goldstone magnons near the M-points remain coherent due to the absence of low-energy two particle states in which to decay (Fig. 5d). This explains the experimental observation of sharp magnon-like modes near the M-points[24]. In contrast, the magnon bands in the remainder of the Brillouin zone directly overlap with the two-particle continuum. It is therefore natural to anticipate a large decay rate even for the lowest magnon bands.

To confirm this idea, we have computed the three-magnon interactions and decay rates for all magnon bands for Model 2 using the self-consistent imaginary Dyson equation (iDE) approach[40]. Within this approach, it is assumed that the real part of the magnon self-energy is already captured by the LSWT parameters, while the imaginary part is obtained self-consistently (see "Methods" and Supplementary Note 3). The iDE approach therefore represents an extension of LSWT, in which the one-magnon excitations are broadened according to the momentum and band-dependent decay rate $\gamma_{\mathbf{k},n}$, while other contributions to the neutron intensity from multi-magnon excitations are also absent[41]. As a result, comparison of LSWT, ED, and iDE results (Fig. 6) allows for the identification of the origin of different contributions to the spectra.

The predicted neutron scattering intensity within the iDE approach (Fig. 6b) captures many of the most notable features that are observed in the ED and experimental data, showing a significant improvement over the LSWT results (Fig. 6a). First, there is an almost complete washout of the two high-energy one-magnon modes due to strong decays. This implies that the higher-energy features >4 meV appearing in ED are primarily multi-magnon in character (including the 6 meV peak at the $\Gamma$-point). The appearance of these higher-energy features in the inelastic neutron response may arise partly from direct contributions from the broadened two-magnon continuum via the longitudinal component of the structure factor, which is not included in the iDE approach (see Supplementary Note 3). Second, the broadening of the two lower magnon bands in the iDE results and the resultant variation of their intensities are in a close agreement with the ED—particularly in a wide region near the $\Gamma$-point (see also Supplementary Fig. 5). These are precisely the features with which the LSWT results were most incompatible. Over much of the Brillouin zone—and especially for the higher magnon bands—the computed $\gamma_{\mathbf{k},n}$ is on the same scale as the one-magnon bandwidth, confirming the absence of coherent magnons.

**Discussion**

The general requirements for strong two-magnon decays are less restrictive than a proximity to a spin liquid state. Indeed, a large decay rate is ensured by the following three conditions: large anisotropic interactions, deviation of the ordered moments away from the high-symmetry axes, and strong overlap of the one-magnon states with the multi-magnon continuum (see Supplementary Note 3). Of these, the first two conditions ensure that the scattering vertex $\Lambda_{12}^3$ is large—of the order of the underlying interactions, i.e., $\Lambda_{12}^3 \sim \mathcal{O}(K_1, \Gamma_1)$. For $\alpha$-RuCl$_3$, the strong overlap with the multi-magnon continuum is ensured by shifting of the low-energy magnons away from the $\Gamma$-point. Since the bottom of the two-magnon continuum must always have an

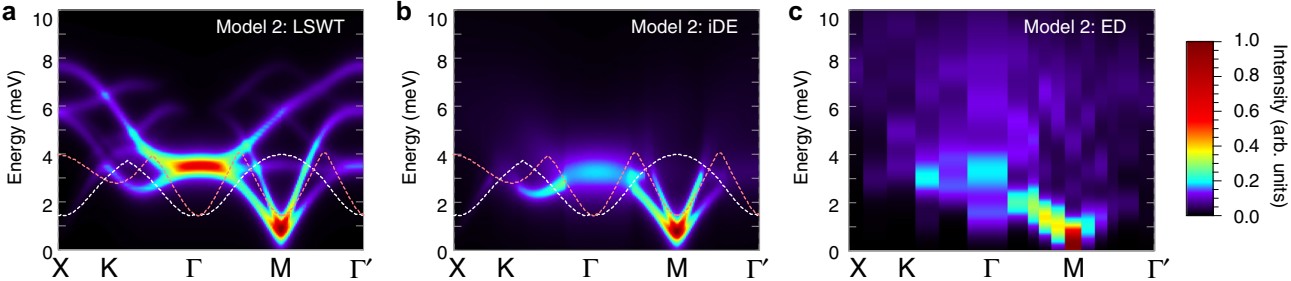

**Fig. 6** Effects of two-magnon decays in $\mathcal{I}(\mathbf{k}, \omega)$ for extended model. Results are shown for Model 2 computed via **a** LSWT, **b** self-consistent iDE approach, and **c** ED. Results in **a** and **b** are averaged over the different zigzag domains. The white and pink dashed lines indicate the bottom of the two-magnon continuum, $E_2^{min}(\mathbf{k})$ for the different zigzag domains. In the iDE results, the effects of two-magnon decays strongly broadens any magnon bands overlapping with the two-magnon continuum

energetic minimum at the $\Gamma$-point, the shifting of the pseudo-Goldstone modes to a finite momentum ensures the remaining higher-energy magnons are degenerate with the continuum near the zone center. Experimentally, these conditions are also likely to be satisfied by the zigzag ordered $Na_2IrO_3$[9], and spiral magnets α-$Li_2IrO_3$, β-$Li_2IrO_3$, and γ-$Li_2IrO_3$[42–44]. This picture is also consistent with recent indications that the magnetically disordered phase observed at high pressure in β-$Li_2IrO_3$[45] is driven primarily by large $\Gamma_1$ interactions[46].

With this in mind, there are two general scenarios that can explain the observed continuum excitations in α-$RuCl_3$ and the iridates $A_2IrO_3$. In the first scenario, which has been advanced by several studies, the excitations can be treated as free particles with a small number of flavors. Such excitations are weakly interacting and have well-defined dispersions, but possess quantum numbers (e.g., $\Delta S_{tot} = \pm 1/2$) or topological properties inconsistent with the experimental neutron scattering selection rules (i.e., $\Delta S_{tot} = 0$, $\pm 1$). The appearance of the broad continuum in energy therefore results only from the fact that these fractional excitations must be created in multiples. If they could have been created individually, they would have represented long-lived and coherent quasiparticles with sharply peaked energies. This scenario indeed describes the Kitaev spin liquid, where the special symmetries of the Hamiltonian allow an exact description in terms of two flavors of particles: non-interacting Majorana spinons and localized fluxes[5]. Such excitations are long-lived, but belong to nontrivial topological sectors, and therefore cannot be created individually by any local operations. For the Kitaev spin liquid, the predicted continuum therefore represents coherent multiparticle excitations.

In contrast, upon moving away from the pure Kitaev point, the relevant symmetries that protect the spinons and fluxes are lifted both by additional magnetic interactions and by spontaneous symmetry breaking of the magnetic order. This tends to confine spinons into gauge neutral objects such as magnons[47, 48]. Despite this latter tendency, we have argued that coherent magnons are unlikely to appear at large $\Gamma_1$ due to the strong anharmonicity in the magnon Hamiltonian. While this leaves open the possibility that nearly free Majorana spinons persist into the zigzag ordered phase in some regions of the Brillouin zone, a more general scenario is that the observed continua represent fully incoherent excitations. In this second scenario, the excitations are not describable in terms of any type of free particles with small decay rates and well-defined dispersions. The broad continua therefore reflect the absence of coherent quasiparticles altogether, rather than particular experimental selection rules related to fractionalization. At present, it is not clear which of these scenarios applies to the iridates and α-$RuCl_3$, although a key role must be played by both the Kitaev $K_1$ and off-diagonal couplings such as $\Gamma_1$. In any case, the study of these materials calls into question the stability of

magnetic quasiparticles in the presence of strongly anisotropic interactions.

In summary, we have shown that all main features of the magnetic excitations in α-$RuCl_3$[12, 13, 24] are consistent with strongly anisotropic interactions having signs and relative magnitudes in agreement with ab initio predictions. The ferromagnetic Kitaev coupling ($K_1 < 0$) is supplemented by a significant off-diagonal term ($\Gamma_1 > 0$) that plays a crucial role in establishing both the zigzag order and the observed continua. In the presence of such interactions, the conventional magnon description breaks down even deep in the ordered phase, due to strong coupling of the one-magnon and two-magnon states. This effect is expected to persist over a large range of the phase diagram suggesting that the observed continua in α-$RuCl_3$ and the iridates $A_2IrO_3$ represent a rich and general phenomenon extending beyond the Kitaev spin liquid. For this class of strongly spin-orbital-coupled magnets, the presence of complex and frustrated anisotropic interactions leads naturally to dominant anharmonic effects in the inelastic magnetic response. Fully describing the dynamics of these and similar materials therefore represents a formidable challenge that is likely to reveal aspects not found in conventional isotropic magnets.

## Methods

**Exact diagonalization**. The neutron scattering intensity was computed via:

$$\mathcal{I}(\mathbf{k}, \omega) \propto f^2(\mathbf{k}) \int dt \sum_{\mu,\nu} \left(\delta_{\mu,\nu} - k_\mu k_\nu / k^2\right) \times \sum_{i,j} \left\langle S_i^\mu(t) S_j^\nu(0) \right\rangle e^{-i\mathbf{k}\cdot(\mathbf{r}_i - \mathbf{r}_j) - i\omega t}, \quad (4)$$

where $f(\mathbf{k})$ is the atomic form factor of $Ru^{3+}$ from ref. [49]. ED calculations were performed using the Lanczos algorithm[50], on several 20-site and 24-site clusters with periodic boundary conditions. Such periodic clusters are detailed in Supplementary Note 4. Excitations were computed using the continued fraction method[51]. Further details and additional results are presented in the Supplementary Notes 4 and 5; these extensive calculations go beyond previous ED studies[16, 17, 20, 26, 29], which focused mainly on the static properties, or a limited portion of the phase diagram. ED results shown for the high-symmetry $\Gamma$, M, Y, X, and $\Gamma'$ points were averaged over all clusters. The ED $\mathbf{k}$-dependence of $\mathcal{I}(\mathbf{k}, \omega)$, integrated over the energy windows $E = 1.3–2.3$, 5.5–8.5, and 10.5+ meV (Figs. 2c, 3c), was obtained from a single 24-site cluster respecting all symmetries of the model. The discrete ED spectra were Gaussian broadened by 0.5 meV, consistent with the width of experimental features[12]. The intensities were also averaged over the same range of out-of-plane momentum as in the experiment[12].

**Linear spin-wave theory**. LSWT results shown in Figs. 1 and 2 were obtained with the aid of SpinW[52]. Following the approach with the ED data, the discrete LSWT spectra were as well Gaussian broadened by 0.5 meV and the intensities were also averaged over the same range of out-of-plane momentum as with ED and in the experiment[12].

**Imaginary self-consistent Dyson equation approach**. In order to calculate magnon decay rates $\gamma_{\mathbf{k},n}$, we have evaluated three-magnon interaction vertices by performing rotation to local reference frames of spins. The obtained value of the real–space interaction is quite large, about ~3 meV. Next, the Born approximation calculation of the decay rates results in unphysical divergencies[34], thus the self-energy $\Sigma_{\mathbf{k},n}$ needs to be regularized. We have used the so-called iDE approach: a

self-consistent solution on the imaginary part of the Dyson's equation, $\Sigma_{\mathbf{k},n}(\epsilon_{\mathbf{k},n} + i\gamma_{\mathbf{k},n}) = -i\gamma_{\mathbf{k},n}$, see ref. [40]. We have obtained the regularized broadening for the magnon spectrum and have calculated the transverse part of the dynamical structure factor, shown in Fig. 6, by adding the calculated decay rates to experimental resolution of 0.25 meV. The spectral function is approximated as a Lorentzian. More technical details can be found in the Supplementary Note 3.

**Code availabilty**. Custom computer codes used in this study are available from the corresponding author upon reasonable request. Documentation of the codes is not available.

**Data availability**. Data are available from the corresponding author upon reasonable request.

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

## Acknowledgements

The authors acknowledge useful discussions with J. Chaloupka, A. Banerjee, S.E. Nagler, A.A. Tsirlin, R. Moessner, F. Pollmann, and M. Zhitomirsky. S.M.W. acknowledges support through an NSERC Canada Postdoctoral Fellowship. R.V. and K.R. acknowledge support by the Deutsche Forschungsgemeinschaft through grant SFB/TR 49. The work of P.A.M. and A.L.C. was supported by the U.S. Department of Energy, Office of Science, Basic Energy Sciences under award no. DE-FG02-04ER46174.

## Author contributions

R.V. and S.M.W. conceived the project. K.R., S.M.W. and A.H. performed and analyzed the ED calculations. P.A.M. and A.L.C. performed and analyzed the iDE results. All authors contributed equally to the manuscript.

## Additional information

**Competing interests:** The authors declare no competing financial interests.

