## [Peer Review File · Nature Communications]

Reviewers' comments:

Reviewer #1 (Remarks to the Author):

This paper reports a theoretical study of magnetic excitations in α -RuCl₃, which is expected to be the closest realization of the Kitaev honeycomb model.

One of the main questions which the authors address in this work is about the nature of the continuum of magnetic excitations which was observed in RuCl₃. While the observed excitation continua in the RuCl₃ has been previously interpreted in terms of signatures of the Kitaev state (Knolle et al), the full description of all observed data in RuCl₃ requires additional interactions beyond Kitaev model, which are thought to be large based on the theoretical analysis performed by this group of authors but also by other theoretical groups. In the presence of large additional interactions, the description of magnetic continuum in terms of fractionalized particles becomes questionable.

The new contribution of the present paper is a detailed study of whether a broad continuum of magnetic excitations can be understood through a conventional magnon decay in a perturbed Kitaev model. In particular, the authors perform a comparative analysis of two different models: the nnHK model and the ab-initio-guided Model 2. The main finding of the paper is that while the nnHK is not sufficient to explain the observed continuum, the ab-initio-guided Model 2 can do it rather well mainly due to the presence of anisotropic interaction Γ which leads to the strong anharmonicity in the magnon Hamiltonian.

The work seems to me thorough and convincing. The results are clearly presented (especially in the Supplemental Materials). Therefore, I think it is potentially suitable for publication in Nature Communications.

However, I have several comments.

1. I did not quite understand what authors mean by "coherent" and "incoherent" continuum. This statement needs a better explanation.
2. Do the authors have some clear explanation why the intensity in the experiment is mainly located for the Γ -point in the momentum space? The explanation provided in the text does not answer to the question "why".
3. It is not completely clear from the paper if both discussed models have pseudo-Goldstone modes at M-points. It seems to me that the answer should be "yes" since both models have zigzag ground state.
4. On page 5 the authors list three conditions for the large magnon decay. Conditions ii) and iii) are not obvious and should be clarified.

Reviewer #2 (Remarks to the Author):

A nicely written paper by S. M. Winter et al. is a very interesting piece of work: it suggests in a rather convincing way that the large continuum of spin excitations, experimentally observed in α -RuCl₃, is *not* due to the onset of the Majorana spinons but rather due to the decays of the magnons — which seems to be inherent to any Kitaev-Heisenberg model with a finite value of the " Γ " interaction. This is a fascinating result which, if indeed fully confirmed [see comment (1) below], definitely deserves a publication in Nature Communications. However, before I can indeed recommend the paper for publication in Nature Comm., I would like to ask the Authors to address the following shortcomings of the paper:

- (1) Actual broadening of spin spectrum due to the magnon decays

The crucial problem that I have with the paper concerns the fact that it is only suggested that the

magnon decays *can* take place in the model. However, the magnon decay rate is not really calculated — not even on the Born approximation level (self-consistent Born would be even better...). In fact, I worry that the calculated broadening of the magnon branches in $S(q,w)$, as resulting from the magnon decays, would not be large enough to account for the observed broadening. For instance, in case of the collinear antiferromagnet in the external magnetic field, see Fig. 9 of <https://journals.aps.org/prb/pdf/10.1103/PhysRevB.82.144402>, the broadening is actually very small. Could it be that in this model it is different?

I strongly encourage the Authors to perform similar calculations as done in several papers by Chernyshev or Zhitomirsky and check whether the broadening is indeed large enough. Only then the claim from the title of the current paper (“ \sim breakdown of magnons in the spin-orbital model for α -RuCl₃”) can be justified.

(2) Agreement with other methods (for model 1)

It would be good if the Authors could compare their results w.r.t. those obtained using the iDMRG method and presented in <https://arxiv.org/pdf/1701.04678.pdf>. It seems to me that the iDMRG results (in particular Fig. 5 and Fig. 8 of the iDMRG paper) do not agree with the presented here ED results of model 1 — even though, in principle, they are obtained, inter alia, for similar ratio of $J/K \sim -0.3$. Could the Authors comment on this discrepancy?

Reviewer #3 (Remarks to the Author):

The paper reports a comprehensive theoretical and calculational investigation of the origins of magnon breakdown in a large and topical class of magnetic systems. The notion of quasiparticles in condensed matter is a powerful and ubiquitous concept that recasts the excited states of a system (which lie behind the dynamical response and transport properties) in terms of a gas of weakly interacting particles. Conversely, understanding the mechanisms by which it breaks down is just as important. In magnetic materials the elementary excitations are spin waves and the particles are magnons; the break down can occur in spin liquids, frustrated systems or ordered magnetic systems with canted moments. Here the focus is on model systems with strong spin-orbit coupling. This has become highly topical recently, with instances of the Kitaev model in iridates such as Na₂IrO₃ and Li₂IrO₃, and most recently α -RuCl₃ in which a continuum of excitations has just been found. While focussing on RuCl₃ and comparison with experimental results published elsewhere, the paper uses this as a vehicle to present a more general analysis of the origin of magnon breakdown in the generalised Kitaev Hamiltonian using the combined approach of linear spin wave theory analysis (LSWT) and exact diagonalization (ED) on small clusters. I believe the work will have a significant impact in the field and I recommend acceptance, subject to an extended discussion of the low energy gap in the excitations in the calculations compared to experimental data (see below).

The first part of the paper shows that the simple Hamiltonian (Model 1) with strong Kitaev $K_1 > 0$ and Heisenberg J_1 used in the analysis of the measured magnetic excitations published in Nature Materials (ref. 13) in a polycrystalline sample is inconsistent with the data — it cannot reproduce the broad continuum and other features found in the most recent data in single crystals — an exploration of parameter space with the combined LSWT and ED simulations are needed for this. They conclude that it is necessary to include additional terms in the Hamiltonian (Eq. 1) and use this Model 2, with exchange parameters inspired by DFT calculations, to show that the all the primary features — continuum and six-fold pattern of diffuse scattering — are reproduced. So far, this is interesting, but would not on its own merit publication in Nature Communications. For me (and I am sure the authors too) the more important part of the paper in the discussion of decay channels for the magnons that follows: it is both pedagogical and novel and the exploration of parameter space of the generalised Kitaev model has a far wider applicability than just this system, drawing out the key importance of the off-diagonal term in the Hamiltonian of Eq 1, Γ_1 , in producing the continuum, rather than the proximity to the spin liquid Kitaev state. This is all elaborated on in the lengthy but valuable supplementary material.

I found the paper to be well written, watertight in its argumentation, novel and with wide applicability to the whole class of strongly spin-orbit coupled model magnetic systems that are so topical.

My only gripe is with the rather cursory reference to the magnitude of the gap that Model 2 (~ 0.5 meV in all the parameter space explored) compared to the measured value of ~ 2 meV (ref 24), which is addressed only in the last paragraph of the supplementary material. They do point out that the gap is influenced by the relative magnitude of K_1 and Γ_1 along three different directions – but given that in ref 24 a pair of values K_1 and Γ_1 is given that do produce the observed gap (albeit within Model 1, with $K_1 < 0$) I think it behoves the authors to explain more carefully how this particular set is inadequate – perhaps a simulation using ED that would show that the continuum is not reproduced (assuming that is the case).

I. SUMMARY OF CHANGES

- Updated the discussion of coherent vs. incoherent excitations on page 11-12 of the manuscript, per the suggestion of Referee 1.
- Updated the discussion of the requirements for strong two-magnon decays on page 10 of the manuscript, per the suggestion of Referee 1.
- Added an additional discussion of requirements for strong two-magnon decays in the supplemental material in response to Referee 1 and 2.
- Added a comment in the main text, and several comments in the supplemental material regarding the magnitude of the gap at the M-point, per the discussion with Referee 3.

II. RESPONSE TO FIRST REFEREE

First Referee: This paper reports a theoretical study of magnetic excitations in α - RuCl_3 , which is expected to be the closest realization of the Kitaev honeycomb model. One of the main questions which the authors address in this work is about the nature of the continuum of magnetic excitations which was observed in RuCl_3 . While the observed excitation continua in the RuCl_3 has been previously interpreted in terms of signatures of the Kitaev state (Knolle et al), the full description of all observed data in RuCl_3 requires additional interactions beyond Kitaev model, which are thought to be large based on the theoretical analysis performed by this group of authors but also by other theoretical groups. In the presence of large additional interactions, the description of magnetic continuum in terms of fractionalized particles becomes questionable.

The new contribution of the present paper is a detailed study of whether a broad continuum of magnetic excitations can be understood through a conventional magnon decay in a perturbed Kitaev model. In particular, the authors perform a comparative analysis of two different models: the nnHK model and the ab-initio-guided Model 2. The main finding of the paper is that while the nnHK is not sufficient to explain the observed continuum, the ab-initio-guided Model 2 can do it rather well mainly due to the presence of anisotropic interaction Γ which leads to the strong anharmonicity in the magnon Hamiltonian.

The work seems to me thorough and convincing. The results are clearly presented (especially in the Supplemental Materials). Therefore, I think it is potentially suitable for publication in Nature Communications.

Authors' Response: We thank the referee for his/her positive comments.

First Referee: However, I have several comments. 1. I did not quite understand what authors mean by "coherent" and "incoherent" continuum. This statement needs a better explanation.

Authors' Response: We have tried to clarify the descriptions in the discussion. It now reads:

- *“There are two general scenarios that can explain the observed continuum excitations in α - RuCl_3 and the iridates A_2IrO_3 . In the first scenario, which has been advanced by several studies, the emergent excitations can be treated as free particles with a small number of flavours: they are weakly interacting, and have well-defined dispersions, but possess quantum numbers (e.g. $\Delta S_{\text{tot}} = \pm 1/2$) or topological properties inconsistent with the experimental neutron scattering selection rules ($\Delta S_{\text{tot}} = 0, \pm 1$). The appearance of the broad continuum in energy therefore results only from the fact that these fractional excitations must be created in multiples experimentally. If they could be created individually, they would represent long-lived and coherent quasiparticles with sharply peaked energies. This scenario indeed describes the Kitaev spin-liquid, where the special symmetries of the Hamiltonian allow an exact description in terms of two flavours of particles: non-interacting Majorana spinons and localized fluxes. Such excitations are long-lived, but belong to nontrivial topological sectors, and therefore cannot be created individually by any local operations. For the Kitaev spin-liquid, the predicted continuum therefore represents coherent multiparticle excitations.*

In contrast, upon moving away from the pure Kitaev point, the relevant symmetries that

protect the spinons and fluxes are lifted both by the additional magnetic interactions, and the spontaneous symmetry breaking of the magnetic order. This tends to confine spinons into gauge neutral objects such as magnons. Despite this latter tendency, we have argued that coherent magnons are unlikely to appear at large Γ_1 , due to the strong anharmonicity in the magnon Hamiltonian. While this leaves open the possibility that nearly free Majorana spinons persist into the zigzag ordered phase in some regions of the Brillouin zone, a more general scenario is that the observed continua represent fully incoherent excitations. In this second scenario, the excitations are not describable in terms of any type of free particles with well defined dispersions. The broad continua therefore reflect the absence of coherent quasiparticles altogether, rather than particular experimental selection rules related to fractionalization. At present, it is not clear which of these scenarios applies to the iridates and α - RuCl_3 , although a key role must be played by both the Kitaev K_1 and off-diagonal couplings such as Γ_1 . In any case, the study of these materials calls into question the stability of magnetic quasiparticles in the presence of strongly anisotropic interactions.”

We hope that this version is more clear.

First Referee: 2. Do the authors have some clear explanation why the intensity in the experiment is mainly located for the Gamma-point in the momentum space? The explanation provided in the text does not answer to the question "why".

Authors' Response: Indeed, we did not directly address this question in the manuscript, because we feel further investigations will be required to make precise statements. There are a few points to consider:

- For the current manuscript, we have limited our investigations to the $(J_1, K_1, \Gamma_1, J_3)$ model at zero temperature. While we show that this model captures many significant aspects of the experimental response for parameters consistent with *ab-initio*, we have certainly not included all symmetry-allowed interactions for simplicity. With further experimental refinement of the model, more precise statements will become possible.
- The relative intensities at the M and Γ -points are highly sensitive to perturbations that affect the gap at the M-point. This is shown, for example, at the level of LSWT:

The left panel shows results for model 2 of the main text, defined by $J_1 = -0.5, K_1 = -5.0, \Gamma_1 = +2.5, J_3 = +0.5$ meV for the k -path shown. We have taken the ordered

wavevector to be $\mathbf{Q} = \mathbf{Y}$, and have not averaged over the different zigzag domains. The next panels show the results for Model 2, modified with a bond-dependency of the magnitude of K_1 and Γ_1 , with signs consistent with the results of Ref. 18 of the main text. Specifically, we show $J_1 = -0.5, J_3 = +0.5$, with $K_1^Z = -5.0 + \delta, K_1^{XY} = -5.0 - \delta, \Gamma_1^Z = +2.5 + \delta/2, \Gamma_1^{XY} = +2.5 - \delta/2$. Since the relative intensities are strongly affected even by small perturbations on the order of $\delta = 0.1K_1$, it is not clear yet how “universal” is the observed dominant intensity at the Γ -point. Perturbations that enhance the magnitude of the gap at the M-point tend to enhance the relative intensity at the Γ -point.

- The experiments in arXiv:1609.00103, which serve as the primary point of comparison, were conducted at $T = 5$ K, which is not necessarily small compared to the ordering temperature of $T_N = 7$ K. At this time, it is not clear (theoretically) what the effects of finite temperature are on the scattering intensity. We are currently working on this aspect, but feel it is beyond the scope of the current manuscript.

In this context, it is worth noting the experimental neutron scattering data reported on α -RuCl₃ so far has largely probed spin-spin correlations with components in the ab -plane. This is true for two reasons. (i) The first, which we have not discussed in the manuscript, is that there is growing evidence for a rather anisotropic g -tensor, with $g_{ab} \gtrsim 2g_{c^*}$ from *ab-initio* and experimental considerations. See, for example, R. Yadav et al., Sci Rep. 2016; 6: 37925. The larger g_{ab} naturally emphasizes correlations in the plane. (ii) The second is: in order to obtain measurements representing the entire 2D Brillouin zone, data presented in e.g. arXiv:1609.00103 and arXiv:1703.01081 was obtained by integrating over finite out-of-plane momenta k_{c^*} for the scattered neutrons. Through the usual geometric factors ($\mathcal{I} \propto \sum(\delta_{\mu,\nu} - k_\mu k_\nu/k^2)\mathcal{S}^{\mu\nu}$), the intensity depends on the spin-spin correlations orthogonal to the scattered momentum, which has only a c^* component at the 2D Γ -point. This effect also emphasizes the ab -plane component of the correlations, particularly near the zone center.

Given this experimental situation, the natural guess is that finite temperature induces some quasi-elastic response centered at wavevectors consistent with the in-plane Curie-Weiss temperature. For Model 2 we have studied, this is given by:

$$\Theta_{ab} = -\frac{1}{4k_B}(3J_1 + 3J_3 + K_1 - \Gamma_1)$$

A large $K_1 < 0$ and $\Gamma_1 > 0$ naturally leads to a ferromagnetic ($\Theta_{ab} > 0$) in-plane Curie-Weiss temperature. Since Θ_{ab} is ferromagnetic (experimentally and theoretically), one would expect the intensity to be mainly located at the 2D Γ -point at higher temperatures. This observation is roughly model-independent, and seems to be consistent with the recent experimental studies above T_N .

Given these observations, understanding the temperature dependence of the scattering intensity and the precise bond-dependencies of the interactions are likely a prerequisite for addressing the referee’s question. We hope the referee agrees that this is beyond the scope of the current manuscript – but these issues are something we are keenly pursuing.

First Referee: 3. It is not completely clear from the paper if both discussed models have pseudo-Goldstone modes at M-points. It seems to me that the answer should be “yes” since

both models have zigzag ground state.

Authors' Response: The referee is correct, the answer is “yes”. This can be seen in Fig. 2(b,c) and Fig. 3(b,c), which show low-energy modes at the M-points. In Model 1, it turns out that there is a hidden continuous (classical) symmetry, which leads to an excitation at the M-point that costs zero energy at the level of LSWT. One might call this a Goldstone mode. In contrast, for Model 2, there is no continuous symmetry, but there are still low-energy modes at the M-points, which we refer to as pseudo-Goldstone modes. Our claim is, for Model 1, that magnon-like excitations (including the Goldstone modes) seem to be stable over most of the Brillouin zone, but in Model 2 *only* the pseudo-Goldstone modes at the M-points are required to remain well-defined.

First Referee: 4. On page 5 the authors list three conditions for the large magnon decay. Conditions ii) and iii) are not obvious and should be clarified.

Authors' Response: We have attempted to clarify this discussion in the main text, and have added another section to the supplemental to address this specifically. The resubmitted version in the main text now reads:

- *“The general requirements for strong two-magnon decays are less restrictive than proximity to a spin-liquid state. Indeed, a large decay rate is ensured by i) large anisotropic interactions, ii) deviation of the ordered moments away from the high-symmetry axes, and iii) the shifting of low-energy magnons away from the Γ -point. Of these, the combination of i) and ii) ensure that the scattering vertex $\Lambda_{\mathbf{12}}^{\mathbf{3}}$ is large – on the order of the underlying interactions, i.e. $\Lambda_{\mathbf{12}}^{\mathbf{3}} \sim \mathcal{O}(K_1, \Gamma_1)$. Condition iii) then suggests large overlap of the one and two-magnon states. Since the bottom of the two-magnon continuum must always have an energetic minimum at the Γ -point, the shifting of pseudo-Goldstone modes to finite momentum ensures the remaining higher energy magnons are degenerate with the continuum near the zone center. Experimentally, these conditions are likely also satisfied by the zigzag ordered Na_2IrO_3 , and spiral magnets α -, β -, and γ - Li_2IrO_3 . This picture is consistent with recent indications that the magnetically disordered phase observed at high pressure in β - Li_2IrO_3 is driven primarily by large Γ_1 interactions.”*

We hope that this version is more clear. We have also provided an extended discussion of these issues in the supplemental material to help further clarify the situation.

III. RESPONSE TO SECOND REFEREE

Second Referee: A nicely written paper by S. M. Winter et al. is a very interesting piece of work: it suggests in a rather convincing way that the large continuum of spin excitations, experimentally observed in α -RuCl₃, is *not* due to the onset of the Majorana spinons but rather due to the decays of the magnons which seems to be inherent to any Kitaev-Heisenberg model with a finite value of the Gamma interaction. This is a fascinating result which, if indeed fully confirmed [see comment (1) below], definitely deserves a publication in Nature Communications. However, before I can indeed recommend the paper for publication in Nature Comm., I would like to ask the Authors to address the following shortcomings of the paper:

Authors' Response: We thank the referee for his/her positive assessment. We address his/her further concerns below.

Second Referee: (1) Actual broadening of spin spectrum due to the magnon decays: The crucial problem that I have with the paper concerns the fact that it is only suggested that the magnon decays *can* take place in the model. However, the magnon decay rate is not really calculated not even on the Born approximation level (self-consistent Born would be even better). In fact, I worry that the calculated broadening of the magnon branches in $S(q,w)$, as resulting from the magnon decays, would not be large enough to account for the observed broadening. For instance, in case of the collinear antiferromagnet in the external magnetic field, see Fig. 9 of [https://journals.aps.org/prb/pdf/10.1103/Phys. Rev. B **82**, 144402](https://journals.aps.org/prb/pdf/10.1103/Phys.Rev.B.82.144402), the broadening is actually very small. Could it be that in this model it is different? I strongly encourage the Authors to perform similar calculations as done in several papers by Chernyshev or Zhitomirsky and check whether the broadening is indeed large enough. Only then the claim from the title of the current paper (breakdown of magnons in the spin-orbital model for alpha-RuCl3) can be justified.

Authors' Response: Since SCBA calculations for magnons are outside our expertise, we have been discussing with experts on these methods, and are currently looking toward pursuing this avenue. However, given that there are several subtleties with these methods (discussed below), we feel that concrete statements can only be made after a complete assessment of various approximations, which is beyond the scope of the current manuscript.

That being said, the evidence for the breakdown of magnons in some of the studied models is already apparent in the exact diagonalization calculations presented in the manuscript and supplemental information. There are clear differences in the frequency and momentum dependence of the computed neutron scattering intensity for e.g. supplemental figures S6(a-

c) and S7(a-c). These are reprinted below for clarity:

While the results of ED and LSWT agree very well for the former case, this is clearly not true for the latter. This observation directly implies the breakdown of magnons (or, in the very least, the dramatic failure of LSWT to describe the excitations in ED calculations on the latter model). We feel it is convincing that a description of the kinetics of the magnon decays allows for the qualitative identification of models where LSWT fails – e.g. the differences between figures S6(a-c) and S7(a-c) are easily understood in this context.

In order to further convince the referee, we can make our argument somewhat more quantitative by estimating the magnitude of the decay rate without making any detailed calculations. We have added a new section to the supplemental material expanding on these ideas. Here, we present a summary. From Rev. Mod. Phys. 85, 219 (2009), within the Born approximation, the magnon scattering rate due to decay into the two-magnon continuum for multiple magnon bands is given by:

$$\gamma(\mathbf{k}, m) = \frac{\pi}{2} \sum_{\mathbf{q}, n, n'} |\Lambda(\mathbf{q}, n, n'; \mathbf{k}, m)|^2 \delta(\epsilon_{\mathbf{k}, m} - \epsilon_{\mathbf{q}, n} - \epsilon_{\mathbf{k}-\mathbf{q}, n'})$$

where $n, n', m \in 1 \dots N$ label the specific magnon band. We can make some simplifying approximations to investigate the relative magnitude of $\gamma(\mathbf{k})$. As suggested in Phys. Rev. B 78, 180413(R) (2008), the vertex $\Lambda(\mathbf{q}, n, n'; \mathbf{k}, m)$ is often a slowly varying function of momentum. Then, it may be sufficient to treat it in some mean-field approximation (i.e. neglect the band and momentum dependence). Then:

$$\gamma(\mathbf{k}, m) \sim \frac{\pi}{2} (\Lambda)^2 g(\epsilon_{\mathbf{k}}, \mathbf{k})$$

where $g(\epsilon_{\mathbf{k}}, \mathbf{k})$ is the two-magnon density of states at the one-magnon energy and momentum $\epsilon_{\mathbf{k}}, \mathbf{k}$. There are two key points regarding Model 2 that we discussed in the manuscript.

- There are no kinematic restrictions to suppress the density of two-magnon states at $\epsilon_{\mathbf{k}}, \mathbf{k}$. Then it is reasonable to expect $g(\epsilon_{\mathbf{k}}, \mathbf{k}) \sim N^2/(2\sqrt{K_1^2 + \Gamma_1^2})$, since the total two-magnon bandwidth is on the order of $2 \times \sqrt{K_1^2 + \Gamma_1^2}$. This is in contrast with conventional antiferromagnetic states at zero field (where the multimagnon states lie above the single one-magnon band, so that $g(\epsilon_{\mathbf{k}}, \mathbf{k}) \sim 0$). The difference directly results from the offset of one- and two-magnon modes in the zigzag phase, as highlighted in the figure below:

(Note that for the study of M. Mourigal et al. (Phys. Rev. B **82**, 144402), the large magnetic field provides both a finite Λ , and finite $g(\epsilon_{\mathbf{k}}, \mathbf{k})$. However, the magnetic field also pushes many of the two-magnon states to higher energies, which may suppress the decay rate.)

- $\Lambda \sim (K_1, \Gamma_1)$ already at zero field for Model 2; the low-symmetry of the interactions and ordered moment direction ensure that the decay vertex is on the same scale as the underlying interactions. This point is now discussed in the updated supplemental material.

Given these conditions, we show in the updated supplemental that the decay rate (e.g. near the Γ -point) for Model 2 would be expected to scale as $\gamma(\mathbf{k}, m) \sim \frac{\pi}{4} N^2 \sqrt{K_1^2 + \Gamma_1^2}$, which is naturally large compared to the single-magnon energy $\epsilon_{\mathbf{k}} \sim \sqrt{K_1^2 + \Gamma_1^2}$. In this sense, there is no reason to expect the single magnon excitations to remain well defined provided they overlap with the continuum. This interpretation is therefore consistent with the results of exact diagonalization, which show predictable departures from LSWT.

Finally, it should be noted that it is not clear that the present models (with $S = 1/2$) can be directly compared to Fig. 9 of M. Mourigal et al. (Phys. Rev. B **82**, 144402), which refers to the $S = 1$ case. The authors of that work note that the situation for the square lattice $S = 1/2$ case is quite different from the higher spin cases. Quoting from Phys. Rev. B **82**, 144402:

- “For the spin-1/2 SAFM the situation appears to be somewhat more delicate. Previous analytical and numerical studies have predicted overdamped one-magnon excitations in a large part of the Brillouin zone. The SCBA scheme used in [M. E. Zhitomirsky and A. L. Chernyshev, Phys. Rev. Lett. **82**, 4536 (1999)] includes a self-consistent renormalization of only one inner magnon line in the decay diagram of Fig. 2 and is, in a sense, not as consistent as the present approach. On the other hand, that approach does take into account the real part of the spectrum renormalization and, as a consequence, the quasiparticle weight redistribution. Such an effect can be deemed small for larger spins but it is more important for $S = 1/2$ and is likely to contribute

to further enhancement of the damping in this case ... Further theoretical efforts may be needed to clarify completely the detailed behavior of the dynamical structure factor for the spin-1/2 SAFM.”

In this sense, a more relevant comparison is therefore Fig. 4 of O. F. Syljuåsen, Phys. Rev. B **78**, 180413(R) (2008) (QMC data), or Fig. 3 of M. E. Zhitomirsky and A. L. Chernyshev, Phys. Rev. Lett. **82**, 4536 (1999) (SCBA approach). One can see that the spectral broadening is much more dramatic for the $S = 1/2$ case than for higher spin values – perhaps due to a large redistribution of spectral weight in addition to a large decay rate (as noted by the authors), or differences in the details of the SCBA approaches. At present, it is not clear how these differences manifest for the strongly anisotropic models considered in our work.

Given that we are not experts on SCBA and similar methods for magnons – and the application of these methods already includes subtleties for simple models – we have chosen to employ exact diagonalization calculations to demonstrate the effects of various terms in the Hamiltonian. We hope that the discussion has nonetheless convinced the referee.

Second Referee: (2) Agreement with other methods (for model 1): It would be good if the Authors could compare their results w.r.t. those obtained using the iDMRG method and presented in <https://arxiv.org/pdf/1701.04678.pdf>. It seems to me that the iDMRG results (in particular Fig. 5 and Fig. 8 of the iDMRG paper) do not agree with the presented here ED results of model 1 even though, in principle, they are obtained, inter alia, for similar ratio of $J/K \sim -0.3$. Could the Authors comment on this discrepancy?

Authors’ Response: We thank the referee for raising this point. In fact, the results from ED and iDMRG are in complete agreement, which validates the results of both approaches (although one has to be careful comparing equivalent frequency ranges and observables). The featuring of Model 1 (with $J_1/K_1 \sim -0.3$) in the main text was specifically motivated by the range of values presented in M. Gohlke et al., arXiv:1701.04678 (our Ref. [33]).

In the iDMRG work, the authors employed $J_1 = \cos \alpha$ and $K_1 = 2 \sin \alpha$, and suggested the model with $\alpha = 0.7\pi$ could reproduce some features of the experimental data. This corresponds to $J_1 \approx -0.6$ and $K_1 \approx 1.6$ in reduced units. In Fig. 5 of the iDMRG paper, the authors show results for $\mathcal{S}(\mathbf{k}, \omega) = \sum_{\mu} S^{\mu\mu}(\mathbf{k}, \omega)$ (with $\mu = \{x, y, z\}$). One should note that this observable differs from the neutron scattering intensity that we have computed (which includes also the form factor and geometric considerations), although we expect their results in Fig. 5 to be comparable to Fig. 2(d) of our paper. M. Gohlke et al. show that (i) There is no response at the 2D Γ -point at an exact frequency of $\omega = 0.4$, (ii) a star-like shape appears at an exact frequency of $\omega = 1.6$, and (iii) a round feature appears at a higher energy $\omega = 2.0$. One should note that the latter energies are, in fact very *high*, on the scale of the Kitaev interaction itself in their calculations.

For Model 1, we chose parameters $J_1 = -2.2$ meV and $K_1 = +7.4$ meV to be consistent roughly with the experimental energy scales. If we rescale the frequencies of M. Gohlke et al. to compare, they roughly correspond to (i) $\omega = 0.4 \rightarrow 1.8$ meV, (ii) $\omega = 1.6 \rightarrow 7.4$ meV, and (iii) $\omega = 2.0 \rightarrow 9.2$ meV. The energy cuts in their figure are therefore essentially consistent with ranges that we have plotted in Fig. 2(d) of the main manuscript. We did this on purpose when making figure 2, since we wanted to facilitate the comparison with the iDMRG results.

In fact, one can see that the results are very consistent. The ED results show no scattering

intensity at the 2D Γ -point at low energies, while the mid-energy range one does see a star-like feature, with additional intensity near the X-points. This is also what is seen in the iDMRG results. The direct comparison can be seen below:

Our main criticism of Model 1 (in reference to α -RuCl₃) is that there are no low-energy excitations at the Γ -point (in contrast to the experiment). The results from iDMRG and ED studies agree on this point. Other small discrepancies between the two methods can be naturally expected, but we believe that they are actually in very good agreement.

IV. RESPONSE TO THIRD REFEREE

Third Referee: The paper reports a comprehensive theoretical and calculational investigation of the origins of magnon breakdown in a large and topical class of magnetic systems. The notion of quasiparticles in condensed matter is a powerful and ubiquitous concept that recasts the excited states of a system (which lie behind the dynamical response and transport properties) in terms of a gas of weakly interacting particles. Conversely, understanding the mechanisms by which it breaks down is just as important. In magnetic materials the elementary excitations are spin waves and the particles are magnons; the break down can occur in spin liquids, frustrated systems or ordered magnetic systems with canted moments. Here the focus is on model systems with strong spin-orbit coupling. This has become highly topical recently, with instances of the Kitaev model in iridates such as Na_2IrO_3 and Li_2IrO_3 , and most recently $\alpha\text{-RuCl}_3$ in which a continuum of excitations has just been found. While focussing on RuCl_3 and comparison with experimental results published elsewhere, the paper uses this as a vehicle to present a more general analysis of the origin of magnon breakdown in the generalised Kitaev Hamiltonian using the combined approach of linear spin wave theory analysis (LSWT) and exact diagonalization (ED) on small clusters. I believe the work will have a significant impact in the field and I recommend acceptance, subject to an extended discussion of the low energy gap in the excitations in the calculations compared to experimental data (see below).

The first part of the paper shows that the simple Hamiltonian (Model 1) with strong Kitaev $K_1 > 0$ and Heisenberg J_1 used in the analysis of the measured magnetic excitations published in Nature Materials (ref. 13) in a polycrystalline sample is inconsistent with the data – it cannot reproduce the broad continuum and other features found in the most recent data in single crystals – an exploration of parameter space with the combined LSWT and ED simulations are needed for this. They conclude that it is necessary to include additional terms in the Hamiltonian (Eq. 1) and use this Model 2, with exchange parameters inspired by DFT calculations, to show that the all the primary features – continuum and six-fold pattern of diffuse scattering – are reproduced. So far, this is interesting, but would not on its own merit publication in Nature Communications. For me (and I am sure the authors too) the more important part of the paper in the discussion of decay channels for the magnons that follows: it is both pedagogical and novel and the exploration of parameter space of the generalised Kitaev model has a far wider applicability than just this system, drawing out the key importance of the off-diagonal term in the Hamiltonian of Eq 1, Γ_1 , in producing the continuum, rather than the proximity to the spin liquid Kitaev state. This is all elaborated on in the lengthy but valuable supplementary material.

I found the paper to be well written, watertight in its argumentation, novel and with wide applicability to the whole class of strongly spin-orbit coupled model magnetic systems that are so topical.

Authors' Response: We thank the referee for his/her positive assessment. Indeed, from our perspective, the recent studies of the so-called Kitaev materials such as Na_2IrO_3 , Li_2IrO_3 , and $\alpha\text{-RuCl}_3$ serve as motivation to reconsider our intuitions regarding the dynamic response of ordered magnetic systems. We feel that some intuitions built from studies of isotropic Heisenberg systems may be seen inappropriate in the context of the strongly anisotropic interactions. To date, most of the experimental studies in the field of “Kitaev materials” have naturally been analyzed with reference to the pure Kitaev model, simply due to the availability of exact results. These previous studies have therefore suggested the observations (such as the breakdown of magnon excitations) to be very specific to the

present class of materials, but we believe them to be more general. We were hoping that the present manuscript would provide an alternative perspective (and language) for discussing these responses in similar anisotropic systems.

Third Referee: My only gripe is with the rather cursory reference to the magnitude of the gap that Model 2 (0.5 meV in all the parameter space explored) compared to the measured value of 2 meV (Ref. 24), which is addressed only in the last paragraph of the supplementary material. They do point out that the gap is influenced by the relative magnitude of K_1 and Γ_1 along three different directions – but given that in Ref. 24 a pair of values K_1 and Γ_1 is given that do produce the observed gap (albeit within Model 1, with $K_1 < 0$) I think it behoves the authors to explain more carefully how this particular set is inadequate – perhaps a simulation using ED that would show that the continuum is not reproduced (assuming that is the case).

Authors’ Response: We have added a sentence in the main text to more clearly point out that our bond-isotropic model does not reproduce the gap at the M-point. This now reads,

- *“The only aspect that is not quantitatively reproduced within the Model 2 is the magnitude of the gap at the M-point (~ 0.8 meV at the level of LSWT vs. ~ 2 meV experimentally). This discrepancy likely results from bond-dependence of the interactions that are allowed within $C2/m$ symmetry, but not considered here for simplicity (see supplemental material).”*

We have also enhanced the discussion of this point in the supplemental material. We provide further comments regarding the model in Ref. 24 below.

Although we have focused on Model 2 in the main manuscript, we conclude the manuscript with a more conservative statement, namely: “The ferromagnetic Kitaev coupling ($K_1 < 0$) is supplemented by a significant off-diagonal term ($\Gamma_1 > 0$) that plays a crucial role in establishing both the zigzag order and the observed continua.” In this sense, the signs of the interactions suggested in Ref. 24 are consistent with our conclusions. In particular: In Ref. 24, the authors analyzed their neutron data using LSWT, and arrived at reasonable fits with a (K_1, Γ_1) -model, with values $K_1 = -6.8$ meV, $\Gamma_1 = +9.5$ meV. They showed that such values can also reproduce the observed spin-wave gap of 2 meV at the level of LSWT.

That being said, we are not completely convinced that the value of Γ_1 is so large, based on our investigations; here are the considerations:

- In the region ($K_1 < 0$) and ($\Gamma_1 > 0$), we have found that the magnitude of the gap at the M-point and relative overall intensities at various k-points in ED calculations is comparable with the results of LSWT (up to finite size effects in the former method). This suggests LSWT may provide reliable clues, even if it does not properly capture the continuum. On this basis, we can make the following observations:
 - The magnitude of the gap at the LSWT level is not strongly sensitive to the ratio of $|K_1/\Gamma_1|$ provided $|K_1/\Gamma_1| > 1$. However, the relative intensities at the various k-points are strongly sensitive to this ratio. We show this below, based on LSWT

results for a single ordering wavevector $\mathbf{Q} = \mathbf{Y}$:

The left panel shows results for model 2 of the main text, defined by $J_1 = -0.5, K_1 = -5.0, \Gamma_1 = +2.5, J_3 = +0.5$ meV for the k -path shown. We have taken the ordered wavevector to be $\mathbf{Q} = \mathbf{Y}$, and have not averaged over the different zigzag domains. The next panels show the results for Model 2, modified by changing the ratio of $|K_1/\Gamma_1|$, as noted. Reproduction of the large experimental intensity at the Γ -point likely requires $|K_1/\Gamma_1| > 1$ in our experience.

- The magnitude of the gap at the LSWT level is strongly sensitive to bond-dependence of the interactions, which do not strongly perturb the rest of the spectrum. This is shown, for example, at the level of LSWT for a single ordering wavevector $\mathbf{Q} = \mathbf{Y}$:

The left panel shows results for model 2 of the main text, defined by $J_1 = -0.5, K_1 = -5.0, \Gamma_1 = +2.5, J_3 = +0.5$ meV for the k -path shown. We have taken the ordered wavevector to be $\mathbf{Q} = \mathbf{Y}$, and have not averaged over the different zigzag domains. The next panels show the results for Model 2, modified with a bond-dependency of the magnitude of K_1 and Γ_1 , with signs consistent with the results of Ref. 18 of the main text. Specifically, we show $J_1 = -0.5, J_3 = +0.5$, with $K_1^Z = -5.0 + \delta, K_1^{XY} = -5.0 - \delta, \Gamma_1^Z = +2.5 + \delta/2, \Gamma_1^{XY} = +2.5 - \delta/2$. The gap can be reproduced already for small perturbations on the order of $\delta = 0.1K_1$, which are within the range suggested by *ab-initio* calculations. At the same time, the large intensity at the Γ -point is retained.

- A key aspect to consider here is that there are two competing phases in the parameter region with $K_1 < 0$ and $\Gamma_1 > 0$. The first is the experimental zigzag phase, which

has low-energy modes at the M-points, and the second is an incommensurate phase with low-energy modes at some incommensurate wavevectors (e.g. between Γ and K). Proximity to the incommensurate phase results in anomalous intensity at low frequencies away from the Γ or M-points in both ED and LSWT calculations.

With this in mind, we have repeated the ED calculations on cluster 24A for the model of Ref. 24, and present the results below. The LSWT results have been now averaged over the three possible zigzag wavevectors, in order to facilitate comparison with the ED results:

The first aspect to point out is that the overall scale of interactions is larger in the model of Ref. 24; we have therefore plotted the intensity cuts from ED calculations in four different energy ranges. While the gap of ~ 2 meV is reproduced at the level of LSWT, this larger gap (compared to our simulations) results partly from a larger *overall* magnitude of the interactions compared to Model 2, and partly from the different ratio of $|K_1/\Gamma_1|$. However, one can see that the low-energy intensity at the Γ -point is suppressed (both in LSWT and ED), while the mid energy modes show anomalously large intensity between Γ and K. These modes are also found in the LSWT results between Γ and K (highlighted with an arrow in the figure above), suggesting proximity to the incommensurate state.

Importantly, there are no experimental observations (to our knowledge) of mid-energy modes near the K-points, which “ruin” the six-fold star shape. This means, experimentally, that the real interactions must be closer to a ferromagnetic state (with low-energy modes at the Γ -point) than the incommensurate state.

On this basis, we are not completely convinced by the model of Ref. 24. That being said, further experimental and theoretical investigations will be required to further refine the interactions in α -RuCl₃. The reason why we have not strongly commented on the size of the gap is that it is likely arising from a combination of Γ_1 and bond-dependencies of the interactions. However, since the gap appears to be quite sensitive to the latter, it does not represent the most direct way to estimate the $|\Gamma_1/K_1|$ ratio.

Reviewers' comments:

Reviewer #1 (Remarks to the Author):

I am happy with the author's reply to my questions. They also did a nice job by revising the manuscript. So, I recommend the manuscript for publication in Nature Communications.

Reviewer #2 (Remarks to the Author):

I am very grateful to the Authors for such a detailed response to my comments. Indeed, it clarifies all of the concerns that I have had (especially concerning the magnon decays). I think that the current version of the paper is in "good shape" and I would like to recommend the paper for publication in Nature Communications.

Reviewer #3 (Remarks to the Author):

The authors have made a number of changes in response to the comments by all three referees.

In my case, the aspect that I requested be addressed was the discrepancy of the magnon gap and experimental data. The authors have added a comment to the main manuscript, and added a figure to the supplementary material together with an enhanced discussion. Further, they have provided yet more detailed discussion in their rebuttal.

I am satisfied with the changes that the authors have made and so recommend acceptance.

I. SUMMARY OF CHANGES

- Added two additional authors: P. A. Maksimov and A. L. Chernyshev, who have contributed explicit estimates of the magnon decay rate, and consequences for the spectra, as discussed by Referee # 2.
- Added a paragraph in the manuscript describing the results obtained with the self-consistent imaginary Dyson equation (iDE) approach, motivated by the suggestions by Referee # 2. This paragraph also includes an additional figure showing a comparison to the neutron scattering intensities calculated with linear spin wave theory and exact diagonalization.

REVIEWERS' COMMENTS:

Reviewer #2 (Remarks to the Author):

I would like to thank the Authors for these additional calculations on magnon decays. Indeed, the presented results show in a convincing way that these are the magnon decays which are mostly responsible for the large continua observable in the spin response of the Kitaev-Heisenberg model calculated using ED. I would just like the Authors to address the following points in the final version of the paper:

(A) Could you please give arguments for the following statements:

"Based on our previous study of the XXZ model [32], we extract the value of $f \sim 1/9$. Back-of-the-envelope estimates suggest this constant to be $f \sim 4/zn^2$ "

In particular, I would appreciate, if you could argue why: (i) f should be similar for the Kitaev-Heisenberg model as for the XXZ model, and (ii) f should be of the order of $4 / zn^2$. Maybe these two points are clear for the experts but unfortunately I cannot easily come up with arguments to support them.

(B) Maybe in the main text of the paper you could mention that, apart from the crucial three-magnon terms and the successive magnon decays, the continuum most probably also arises from the longitudinal response (as shown in the bottom-most panel of Fig. S5).

Altogether, in my opinion, the additional material provided by the Authors is of excellent quality. Crucially, the presented results confirm the main conclusions of the paper and further support my decision from few weeks ago to recommend the paper for publication in Nature Communications.

I. RESPONSE TO REFEREE #2

Referee #2: *I would like to thank the Authors for these additional calculations on magnon decays. Indeed, the presented results show in a convincing way that these are the magnon decays which are mostly responsible for the large continua observable in the spin response of the Kitaev-Heisenberg model calculated using ED. . . . Altogether, in my opinion, the additional material provided by the Authors is of excellent quality. Crucially, the presented results confirm the main conclusions of the paper and further support my decision from few weeks ago to recommend the paper for publication in Nature Communications.*

Authors' Response: We are very thankful to the referee for such a generous endorsement of our results.

Referee #2: *I would just like the Authors to address the following points in the final version of the paper:*

(A) *Could you please give arguments for the following statements:*

"Based on our previous study of the XXZ model [32], we extract the value of $f \sim 1/9$. Back-of-the-envelope estimates suggest this constant to be $f \sim 4/zn^2$ "

In particular, I would appreciate, if you could argue why: (i) f should be similar for the Kitaev-Heisenberg model as for the XXZ model, and (ii) f should be of the order of $4 \sim zn^2$. Maybe these two points are clear for the experts but unfortunately I cannot easily come up with arguments to support them.

Authors' Response: We admit that the explanation of the matter was indeed a bit brief. We modify it by providing the following paragraph after Eq. (20) of the Supplemental:

- *"Back-of-the-envelope estimates suggest this constant to be $f \simeq 4/zn^2$, where z is the coordination number and n is the number of magnon branches (sites in the unit cell). This estimate comes from analyzing the structure of the dimensionless cubic vertices $\tilde{\Phi}_{\mathbf{q},\mathbf{k}-\mathbf{q};\mathbf{k}}^{\nu\mu}$ in previous studies such as Supplementary Ref. [30]. As is clear from Supplementary Eq. (14), the real-space coupling of spin-flips affects nearest-neighbour Holstein-Primakoff magnons. Hence, the vertex in \mathbf{k} -space contains an analog of the nearest-neighbour hopping matrix. Averaging of its square yields with $\sim 1/z$ the inverse*

coordination number. The number of atoms in the magnetic unit cell n gives the number of independent magnon modes and, therefore, their wavefunctions are normalized by $1/\sqrt{n}$. Since the vertex couples three magnons, its square is, thus, proportional to $\sim 1/n^3$. The summation over such modes eliminates one power of n . The factor of 4 comes from the square of the symmetrization factor in the decay term. For the considered problem of $n = 4$ and $z = 3$ the value of $f \simeq 1/12$ is in a quantitatively close agreement with the value of $f \simeq 1/9$. ”

We believe that this addresses both points raised by referee.

Referee #2: (B) *Maybe in the main text of the paper you could mention that, apart from the crucial three-magnon terms and the successive magnon decays, the continuum most probably also arises from the longitudinal response (as shown in the bottom-most panel of Fig. S5).*

Authors’ Response: We have modified the discussion to include this point. It now reads:

- *“This implies that the higher-energy features > 4 meV appearing in ED are primarily multi-magnon in character (including the 6 meV peak at the Γ -point). The appearance of these higher energy features in the inelastic neutron response may arise partly from direct contributions from the broadened two-magnon continuum via the longitudinal component of the structure factor, which is not included in the iDE approach (see Supplementary Note 3)”*